# Assessment of Inspiratory Effort in Spontaneously Breathing COVID-19 ARDS Patients Undergoing Helmet CPAP: A Comparison between Esophageal, Transdiaphragmatic and Central Venous Pressure Swing

**DOI:** 10.3390/diagnostics13111965

**Published:** 2023-06-05

**Authors:** Sergio Lassola, Sara Miori, Andrea Sanna, Ilaria Menegoni, Silvia De Rosa, Giacomo Bellani, Michele Umbrello

**Affiliations:** 1Anesthesia and Intensive Care 1, Santa Chiara Hospital, APSS Trento, 38122 Trento, Italy; sergio.lassola@apss.tn.it (S.L.);; 2Centre for Medical Sciences—CISMed, University of Trento, Via S. Maria Maddalena 1, 38122 Trento, Italy; 3Anesthesia and Intensive Care 2, San Carlo Borromeo Hospital, ASST Santi Paolo e Carlo—Polo Universitario, 20148 Milano, Italy

**Keywords:** COVID-19, ARDS, esophageal pressure, central venous pressure, inspiratory effort

## Abstract

Introduction: The clinical features of COVID-19 are highly variable. It has been speculated that the progression across COVID-19 may be triggered by excessive inspiratory drive activation. The aim of the present study was to assess whether the tidal swing in central venous pressure (ΔCVP) is a reliable estimate of inspiratory effort. Methods: Thirty critically ill patients with COVID-19 ARDS underwent a PEEP trial (0–5–10 cmH_2_O) during helmet CPAP. Esophageal (ΔPes) and transdiaphragmatic (ΔPdi) pressure swings were measured as indices of inspiratory effort. ΔCVP was assessed via a standard venous catheter. A low and a high inspiratory effort were defined as ΔPes ≤ 10 and  >15 cmH2O, respectively. Results: During the PEEP trial, no significant changes in ΔPes (11 [6–16] vs. 11 [7–15] vs. 12 [8–16] cmH2O, p = 0.652) and in ΔCVP (12 [7–17] vs. 11.5 [7–16] vs. 11.5 [8–15] cmH2O, *p* = 0.918) were detected. ΔCVP was significantly associated with ΔPes (marginal R^2^ 0.87, *p* < 0.001). ΔCVP recognized both low (AUC-ROC curve 0.89 [0.84–0.96]) and high inspiratory efforts (AUC-ROC curve 0.98 [0.96–1]). Conclusions: ΔCVP is an easily available a reliable surrogate of ΔPes and can detect a low or a high inspiratory effort. This study provides a useful bedside tool to monitor the inspiratory effort of spontaneously breathing COVID-19 patients.

## 1. Introduction

The clinical features of COVID-19 are highly variable and may change remarkably with time. It has been speculated that disease progression may be triggered by excessive inspiratory drive activation [1] and that the magnitude of inspiratory effort may be correlated with the need to switch to invasive ventilation [2,3].

Non-invasive oxygenation strategies help preserve physiological protective airway reflexes [4,5] and may directly reduce the complications of endotracheal intubation and invasive mechanical ventilation [5,6,7]. On the other hand, recent studies have suggested that spontaneous breathing could also represent a potential mechanism of lung injury if acute respiratory distress is severe [8]. Strong inspiratory efforts may produce uncontrolled tidal changes in dynamic transpulmonary pressure, thus generating the onset of self-inflicted lung injury (SILI) [9,10].

Esophageal pressure (Pes) represents the gold standard technique to measure the pressure generated by respiratory muscles [11,12] and to calculate the pressure-time product (PTP) and the work of breathing (WOB). However, such measures require an esophageal catheter, which is not routinely used in the clinical setting [13]. The COVID-19 pandemic created an imbalance between ICU beds, care needs, and available resources [14], thus highlighting the need for bedside-available tools for the assessment of patient respiratory mechanics in everyday clinical practice.

The superior vena cava is an intrathoracic vein characterized by high compliance. This feature explains the impact of intrathoracic pressure variation, during both mechanical and spontaneous ventilation, on central venous pressure (CVP) values. Therefore, based on these physiological assumptions, the tidal swing in central venous pressure (ΔCVP) may reflect intrathoracic (i.e., pleural) pressure change, which can be estimated from the tidal swing in esophageal pressure (ΔPes) [15,16,17]. However, limited evidence still exists regarding the use of ΔCVP to assess patient inspiratory effort.

The aim of the present study is to assess whether a bedside-available index such as the tidal swing in central venous pressure (ΔCVP) can reliably predict inspiratory effort, as assessed by esophageal pressure swing (ΔPes) and transdiaphragmatic pressure (ΔPdi), in a cohort of consecutive patients with COVID-19 acute respiratory distress syndrome (C-ARDS) during spontaneous breathing with helmet CPAP (Continuous Positive Airway Pressure). Secondary outcomes are the diagnostic capability of ΔCVP to detect “high” or “low” inspiratory effort, defined by specific thresholds of ΔPes, and the effect of varying the levels of PEEP on inspiratory effort.

## 2. Materials and Methods

### 2.1. Subjects

This prospective observational cohort study was performed in the ICU at Santa Chiara Hospital, Trento, Italy, between March and December 2021.

Ethical approval for this study (Rep. Int.282/2022) was provided by the Comitato Etico per le Sperimentazioni Cliniche of the Azienda Provinciale per i Servizi Sanitari di Trento (Chairperson prof. Giuseppe Moretto) on 24 February 2022; written informed consent was obtained according to Italian regulations, and all procedures were in accordance with the Helsinki Declaration of 1975.

Inclusion criteria were: age greater than 18 years, the diagnosis of COVID-19 acute respiratory failure with a PaO_2_/FiO_2_ ratio < 200 mmHg, the presence of spontaneous breathing during helmet CPAP respiratory support, and enrolment within 48 h of ICU admission. Exclusion criteria were: acute cardiogenic pulmonary edema, massive pulmonary embolism, COPD, interstitial lung disease, neuromuscular disease, chest-wall deformities, neurological deterioration, pregnancy, respiratory or cardiopulmonary arrest, helmet CPAP intolerance or refusal to participate.

### 2.2. General Measures

Demographic information and relevant comorbidities were recorded on admission. The Simplified Acute Physiology Score, the Sequential Organ Failure Assessment score, arterial blood gases (PaO_2_ and PaCO_2_), pH, PaO_2_/FiO_2_ ratio, blood lactate values, respiratory rate (RR), SpO_2_, arterial pressure, heart rate were assessed and recorded on admission and before helmet CPAP. At admission, patients underwent a diaphragm ultrasound to assess the inspiratory thickening fraction of the muscle, as previously reported [18]. Diaphragm thickness was assessed in the zone of apposition of the diaphragm to the ribcage. A 12-Mhz linear probe was placed above the right 10th rib in the mid-axillary line, and the thickening fraction (TF) was calculated as the ratio between the difference of inspiratory and expiratory thickness, divided by the expiratory thickness of the diaphragm.

### 2.3. Physiological Measurements

A nasogastric tube (NutriVent^®^, SIDAM, Mirandola, Modena, Italy) provided with two polyethylene balloons suitable for registering the esophageal (Pes) and gastric (Pga) pressures was introduced into the stomach as per manufacturer instructions. Data were sampled at 100 Hz and processed on a dedicated data acquisition system (Optivent^®^, SIDAM, Mirandola, Modena, Italy). The appropriate position of the esophageal balloon was evaluated by monitoring the presence of the cardiac artifact on the esophageal pressure tracing. The maximal negative deflection of the esophageal pressure from the end-expiratory value was defined as esophageal pressure swing (ΔPes) [19]. Transdiaphragmatic pressure (Pdi) represents the pressure across the diaphragm, and it is measured from simultaneous differences between Pes and Pga. ΔPdi is the inspiratory swing in Pdi [20,21]. All measurements were performed in the semi-recumbent position during a 5 min stable spontaneous breathing pattern, and results were averaged for each assessment step. The onset of inspiratory effort was identified at the moment of the initial decay of Pes.

Central venous pressure (CVP) was measured through the distal port of a central venous catheter (triple lumen), with the distal end in the superior vena cava assessed by chest X-ray. The CVP measurement was zeroed at mid-chest at the level of the fifth rib [22], and the value was taken at the base of the “c” wave, either at the end of inspiration or at the end of expiration [23]. CVP swing (ΔCVP) was calculated as the maximal negative deflection of the CVP from the end-expiratory value. The difference between CVP and Pes (transmural CVP) was calculated [24]. CVP trace was recorded on a dedicated multiparametric monitor (Carescape B850, GE Healthcare, Little Chalfont, UK) for subsequent measurements.

### 2.4. Study Protocol

After ICU admission, non-invasive respiratory assistance with helmet CPAP with a Venturi flow generator (Dimar, Mirandola, Modena, Italy) was started with gas flow 100 L/min and FiO_2_ level to reach SpO_2_ > 92%. When stable, patients underwent a trial with three increasing levels of PEEP, lasting 30 min each. The first level was set at 0 cmH_2_O(ZEEP); PEEP was then increased to 5 (PEEP5) and 10 cmH_2_O (PEEP10); FiO_2_ did not change. The pattern of breathing, Borg scale dyspnea score, indices of respiratory effort, and hemodynamic parameters were recorded in the last 5 min of each step. Sedation could be administered to ensure patient comfort; the level of sedation (SAS scale) and the sedatives used were recorded. The protocol could be stopped if the patient showed signs of respiratory distress (respiratory rate > 35 breaths/min, SpO_2_ < 90%, diaphoresis or anxiety, heart rate >140 bpm, systolic blood pressure > 180 mmHg).

### 2.5. Statistical Analysis

Data were analyzed using R Core Team 2022 (R: A language and environment for statistical computing. Version 1.13. R Foundation for Statistical Computing, Vienna, Austria. https://R-project.org/ (accessed on 22 April 2022).

Based on the data from a previous investigation carried out in mechanically-ventilated patients with C-ARDS, in which the average coefficient of determination for the relationship between ΔCVP and ΔPes was 0.8 [25], and hypothesizing that such coefficient could be at least 10% higher in spontaneously breathing patients, we calculated that a sample size of 30 subjects would be required to detect a significant correlation (80% power and alpha = 0.05)

The Shapiro-Francia test was used to investigate the normality. Data were expressed as mean and standard deviation for normal distributions or median and interquartile range (IQR) for all other cases. The student’s *t*-test, the Mann-Whitney rank-sum test, and Fisher’s Exact test were used to compare variables of interest between groups.

The analysis of variance for repeated measurements was used to analyze the variables recorded over the three steps (ZEEP, PEEP5, PEEP10), with a step as a within-subject factor, and the Greenhouse–Geisser method was used to correct the statistical significance of the within-subject factors. The Friedman test was used to analyze non-parametric variables. Pairwise, post-hoc multiple comparisons were carried out according to the Tukey method.

The correlation between ΔCVP and ΔPes was investigated by a linear mixed model for repeated measures to deal with the longitudinal structure of our data set. The coefficient of determination for the mixed model (marginal R^2^) and *p*-value were used to express the association between variables.

The diagnostic performance of ΔCVP to detect low and high inspiratory effort (defined in accordance with the literature as an esophageal pressure swing ≤ 10 and >15 cmH_2_O, respectively) [26,27,28] was assessed by the area under the Receiver Operating Characteristic (ROC) curve, sensitivity, specificity, positive and negative predictive value. The best cut-off point was the value with the highest sensitivity and specificity (Youden’s Index). Two-tailed *p*-values  <  0.05 were considered for statistical significance.

## 3. Results

### 3.1. Patient Characteristics

Thirty consecutive patients were included in the study. The characteristics of the patients at baseline are reported in Table 1. Patients were studied on average 1 (0–1) days after ICU admission. Twenty-five patients (83.4%) were admitted from a medical ward, and five (16.6%) from the emergency department. Twenty-three patients (76.7%) received a low-dose continuous infusion of remifentanil to increase comfort and compliance with the medical devices; the Riker Sedation-Agitation Scale (SAS) was 4 (4; 4).

### 3.2. Effects of Different PEEP Levels during Helmet CPAP on Respiratory Mechanics and Hemodynamic Parameters

Table 2 shows hemodynamic parameters and respiratory effort values during the three levels of the CPAP trial. No significant changes in MAP were detected, while heart rate and respiratory rate were significantly reduced, and SpO_2_ increased. No significant changes in esophageal (ΔPes) and central venous pressure swing (ΔCVP) were noted, while ΔPdi progressively decreased. No significant changes in the Borg dyspnea scale were detected during the study.

### 3.3. ΔCVP-ΔPes Relationship

There was a significant correlation between ΔCVP and ΔPes (marginal R^2^ = 0.87, *p* < 0.001), as described in Figure 1.

Figure 2 shows the effect of the PEEP level on the degree of correlation between ΔCVP and ΔPes. The correlation was significant during the three steps of the study: r^2^ = 0.92 (*p* < 0.001) at ZEEP, r^2^ = 0.89 (*p* < 0.001) at PEEP 5, r^2^ = 0.83 (*p* < 0.001) at PEEP 10.

### 3.4. Diagnostic Performance of ΔCVP to Detect Low and High Inspiratory Effort

The area under the ROC curve for the detection of a low inspiratory effort was 0.899 (95% CI: 0.838–0.960) for ΔCVP (Figure 3, right panel), with the best cut-off of 11 cmH_2_O. The area under the ROC curve for the detection of a high inspiratory effort was 0.982 (95% CI: 0.961–1) (Figure 3, left panel), with the best cut-off value of 15 cmH_2_O. Table 3 shows the diagnostic performance of the best cutoff values for central venous pressure swing to detect low and high inspiratory effort.

## 4. Discussion

The main findings of this study are: (1) in spontaneously breathing patients with helmet CPAP, bedside-available ΔCVP is significantly associated with the level of patient inspiratory effort as assessed by the ΔPes value; (2) ΔCVP is an easily available and reliable surrogate of ΔPes for detecting low or dangerous inspiratory effort, as defined by specific thresholds of esophageal pressure, (3) in the early phase of C-ARDS, ΔPes, and ΔCVP do not change with the increase of the PEEP levels.

### 4.1. Physiological Characteristics of a Cohort C-ARDS Patients

This study examined C-ARDS features in a cohort of thirty consecutive spontaneously breathing patients with helmet CPAP within 48 h of ICU admission. In particular, we measured the effects of the different PEEP levels on physiological parameters (Table 2).

During the PEEP trial, there was no significant change in esophageal pressure swing and central venous pressure swing, possibly due to the fact that lung compliance is relatively preserved at the beginning of the disease [29]. This differs from the decrease in esophageal pressure swing after elevated PEEP levels, observed in typical recruitable ARDS [9,30,31]. In a study on 140 COVID-19 patients, Coppola and co-workers [32] were not able to find any change in esophageal pressure swing during the PEEP trial (0 and 10 cmH_2_0). Tonelli and co-workers [33] also recently found, in 30 COVID-19 patients, esophageal pressure swing values comparable to our population. The authors described that the magnitude of inspiratory effort in patients with COVID-19 was about 50% lower than in patients without COVID-19 acute respiratory distress syndrome and that the significance of self-inflicted lung injury was probably limited. In particular, a relatively normal respiratory drive may explain the ‘happy hypoxemia’ observed in our and other populations [34] and the almost complete absence of dyspnea as assessed by the Borg dyspnea scale.

As for the transdiaphragmatic pressure, we found a reduction of ΔPdi values during the steps of the study. Indeed, since the diaphragm is a curved surface, the pressure difference across it is proportional to the tension of the muscle and inversely proportional to the radius of curvature of the diaphragm (Laplace’s law) [35]. This means that increasing the radius of curvature of the diaphragm reduces its ability to generate pressure, as it happens when the PEEP level increases [36,37,38]. However, the diaphragm contractile capacity of our patients was preserved, as assessed by the values of the diaphragm thickening ratio measured at the time of ICU admission.

Moreover, after positioning the helmet CPAP, the respiratory rate was significantly reduced, even if it was not influenced by the PEEP values. On the other hand, SpO_2_ values progressively improved with increasing PEEP, possibly due to the redistribution of the lung perfusion [39].

### 4.2. Inspiratory Effort and Central Venous Pressure Swing

When the inspiratory muscles contract, there is a progressive reduction of the pleural pressure values. Its variation can be estimated by the variation of the esophageal pressure at the bedside [20,40]. The reference methods for measuring inspiratory effort are the work of breathing (WOB) and the pressure-time product (PTP). The first analyzes the variation of the muscular pressure (Pmusc) with respect to the generated volume using the Campbell diagram. The PTP, on the other hand, analyzes the variation of muscular pressure with respect to the duration of inspiration [41]. In clinical practice, the tidal swing in esophageal pressure (ΔPes) is probably sufficient to monitor inspiratory effort [42].

The superior vena cava has intrinsic elastic properties with high compliance. This feature allows the transmission of the pleural pressure variation to the vascular structures. Therefore, the central venous catheter could be used to estimate the inspiratory effort; although it is well known that CVP and Pes fluctuate in a similar way [43], the use of ΔCVP to estimate inspiratory effort is still poorly implemented in clinical practice. In a recent editorial regarding COVID-19 patients, the use of CVP swings as a surrogate measure for the work of breathing was recommended [29]. In our population, we found that ΔCVP was an adequate estimate of inspiratory effort as assessed by ΔPes, which provides a global assessment of all inspiratory muscles [20,40,42], whereas ΔPdi is specific to the diaphragm [20]. As a result, ΔCVP is likely to better reflect the fall in intrathoracic pressure generated by all inspiratory muscles and not just the effect of diaphragmatic contraction.

Similar results were reported by other authors. Chieveley-Williams et al. found that the ratio of ΔCVP to ΔPes was on average 1.1, with a mean difference of 1 cmH_2_O when the level of pressure support was modified in 10 patients undergoing pressure support ventilation [17]. Colombo et al. found that, in 48 critically-ill subjects breathing spontaneously with zero end-expiratory pressure or 10 cmH_2_O of PEEP, ΔCVP correlated with ΔPes, with a stronger relationship at 0 PEEP than 10 PEEP (R^2^ 0.81 vs. 0.67) [28]. Recently, we demonstrated that in 14 patients with C-ARDS during a descending trial with three levels of pressure support ventilation, the esophageal and central venous pressure tidal swing was significantly associated (R^2^ = 0.810, *p* < 0.001) [25].

Different factors may affect the concordance between Pes and CVP. In particular, two different systems are compared: the central venous catheter is filled with fluid, while the esophageal balloon is filled with air [44]. Moreover, the patient volume status could heavily influence the measurement of tidal ∆CVP. Indeed an increase in the filling state of the vein could lead to a reduction in its compliance and, therefore, in the transmission of the variation in pleural pressure to the vascular structure [15].

### 4.3. Diagnostic Performance

We investigated the ability of central venous pressure swing (ΔCVP) to identify a high or a low inspiratory effort with respect to defined threshold values of esophageal pressure swing (ΔPes) [26,27,28]. In particular, ΔCVP recognized low and high inspiratory efforts with excellent diagnostic performance: ΔCVP < 11 cmH_2_O rules in a low inspiratory effort, whereas ΔCVP > 15 cmH_2_O rules in a high inspiratory effort.

### 4.4. Clinical Consequences

In this study, we showed how the inspiratory effort could be identified by tidal central venous pressure swing with acceptable accuracy. Since excessive inspiratory effort can lead to complications such as self-inflicted-lung-injury (SILI) [45,46], ΔCVP could be a useful tool to track the dangerous inspiratory effort in patients with C-ARDS, especially in the case of limited resources. Moreover, ΔCVP could be used to screen patients who would eventually benefit from monitoring Pes [47].

### 4.5. Limitations

The limitations of this study are: First, we studied a small population which may limit the generalization of our results. However, our population is comparable to other physiological studies [2,33]. Second, the current investigation, estimated the inspiratory effort by esophageal pressure swing (ΔPes) even though the gold standard measurement is PTP. Third, in a non-compliant heart, the transmission of pleural pressure to the cardiac cavities can be blunted, and even small changes in volume can produce large changes in pressure: in such cases, the agreement between ΔCVP and ΔPes will be particularly poor [47,48], so we caution against the extension of our results to patients with significant cardiac pathology. Further studies are warranted to verify if our findings could be reproduced in non-COVID-19-related acute respiratory failure patients.

## 5. Conclusions

Central venous pressure swing (ΔCVP) is a bedside-available tool to track the inspiratory effort in spontaneously breathing COVID-19 ARDS patients undergoing helmet CPAP.

## Figures and Tables

**Figure 1 diagnostics-13-01965-f001:**
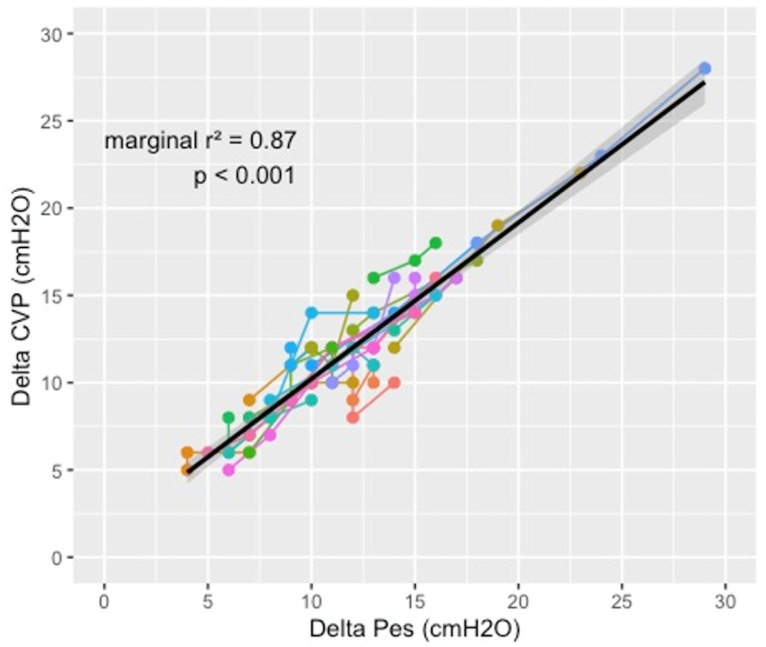
Relationship between central venous pressure swing and esophageal pressure swing during the different phases of the study. Each color represents a patient; the dots, connected by a line, represent the three measures at different PEEP levels. The solid lines represent the linear predictions, while the grey area is their 95% confidence interval. Analysis was performed for all patients using a mixed model for repeated measures to account for the longitudinal structure of our data set (patients with repeated measurements over time).

**Figure 2 diagnostics-13-01965-f002:**
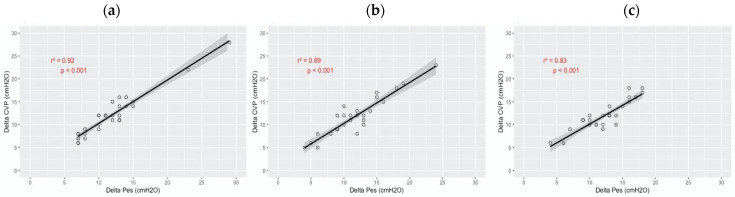
Correlation of central venous pressure swing and esophageal pressure swing during the different phases of the study. The solid lines represent the linear predictions, while the gray area is their 95% confidence interval. The analysis was performed by linear regression. (**a**) ZEEP. (**b**) PEEP 5. (**c**) PEEP 10.

**Figure 3 diagnostics-13-01965-f003:**
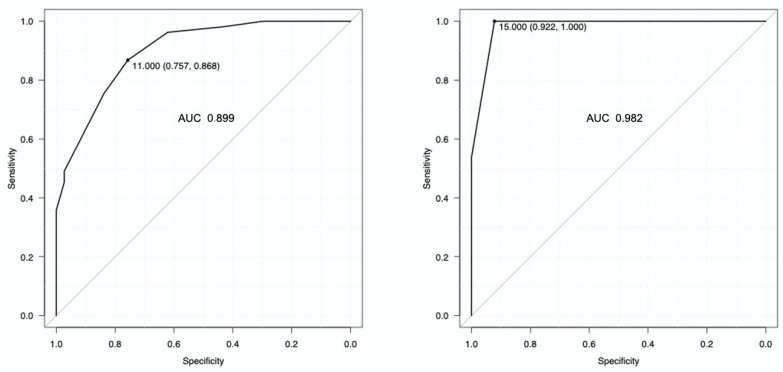
Diagnostic performance of the central venous pressure swing to detect low inspiratory effort (**right panel**) and high inspiratory effort (**left panel**), defined as an esophageal pressure swing ≤ 10 and >15 cmH_2_O, respectively. The best cut-off for detecting a low inspiratory effort was 11 cmH_2_O, while the best cut-off for detecting a high inspiratory effort was 15 cmH_2_O.

**Table 1 diagnostics-13-01965-t001:** Clinical characteristics and outcomes.

	*N* = 30
**Anthropometric measures**	
Male Sex (*n*—%).	18 (60)
BMI (kg/m^2^)	29.5 ± 6.9
Age (years)	65.6 ± 9.8
**Comorbidities**	
Hypertension (*n*—%)	17 (60.0)
Diabetes (*n*—%)	5 (16.7)
Cancer (*n*—%)	2 (6.7)
Cardiac failure (*n*—%)	1 (3.3)
Respiratory disease (*n*—%)	4 (13.3)
Immunosuppression (*n*—%)	4 (13.3)
**ICU severity scores**	
SAPS II	30.8 ± 5.9
SOFA	3.6 ± 1.1
**Parameters at enrolment**	
Temperature (°C)	36.6 ± 0.6
FiO_2_	0.61 ± 0.14
PaO_2_/FiO_2_ (mmHg)	128.0 ± 45.9
PaCO_2_ (mmHg)	40.0 ± 5.6
PaO_2_ (mmHg)	75 ± 20
pH	7.41 ± 0.05
Respiratory Rate (1/min)	22.5 ± 4.9
Diaphragm thickening ratio (%)	29.6 ± 17.3
**Outcome**	
Endotracheal intubation (*n*—%)	20 (66.7)
ICU deaths (*n*—%)	10 (33.3)
ICU length of stay (days)	16 (8–29)
Hospital length of stay (days)	29 (18–36)

N sample size, BMI body mass index, SAPS II simplified acute physiology score, 2nd version, SOFA sequential organ failure assessment score, FiO_2_ fraction of inspired oxygen, ICU intensive care unit.

**Table 2 diagnostics-13-01965-t002:** Hemodynamic parameters and respiratory effort indices at different PEEP levels.

Variable	PEEP 0	PEEP 5	PEEP 10	*p* Value
Heart rate (bpm)	79 ± 13.3	76.5 ± 12.6 °	76 ± 13.6 °	<0.001
Mean arterial blood pressure (mmHg)	93.5 ± 12.5	93.5 ± 12.7	93.7 ± 12.5	0.871
Transmural vascular pressure (mmHg)	3 ± 5.9	2 ± 6.1 °	2 ± 6.1 °	<0.001
Respiratory rate (bpm)	22.5 ± 4.9	19.5 ± 4.5 °	19 ± 4.2 °	0.006
SpO_2_ (%)	91 ± 4.2	93 ± 3.6 °	94.5 ± 3.1 °*	<0.001
Pes, exp (cmH_2_O)	8.1 ± 3.7	11.1 ± 4.3 °	13 ± 4.3 °	<0.001
Pes, insp (cmH_2_O)	−3.6 ± 5.9	−0.4 ± 5.6 °	0.9 ± 5.7 °	0.010
ΔPes (cmH_2_O)	11 ± 4.8	11 ± 4.3	12 ± 3.7	0.652
CVP, exp (cmH_2_O)	11.5 ± 5.2	12.6 ± 6.3	14.2 ± 5.7	0.177
CVP, insp (cmH_2_O)	−0.4 ± 6	1 ± 5.9	2.4 ± 6.2	0.209
ΔCVP (cmH_2_O)	12 ± 4.7	11.5 ± 4.1	11.5 ± 3.3	0.918
Transdiaphragmatic pressure (cmH_2_O)	18 ± 6.6	16.5 ± 5.9 °	14 ± 6.3 °	0.003
Borg scale	1 (0–2)	1 (0–2)	2 (1–2.5)	0.222

Data are expressed as mean ± standard deviation. ΔPes esophageal pressure swing, ΔCVP central venous pressure swing, exp expiratory, insp inspiratory. Analysis of variables recorded at three different steps (ZEEP, PEEP 5, and PEEP 10) was performed for all patients using analysis of variance for repeated measurements with a step as a within-subject factor in the case of normally-distributed variables. The significance of the within-subject factors was adjusted using the Greenhouse-Gaisser method. Non-parametric variables were analyzed using the Friedman test. Pairwise post-hoc multiple comparisons were performed if necessary. ° *p* < 0.05 compared with ZEEP; * *p* < 0.05 compared with PEEP 5.

**Table 3 diagnostics-13-01965-t003:** Diagnostic performance of the best cut-off values for central venous pressure swing to detect low and high inspiratory effort.

	Sensitivity [95% CI]	Specificity [95% CI]	PPV [95% CI]	NPV [95% CI]
Low inspiratory effort				
ΔCVP < 11 cmH_2_O	86.8 [74.7–94.5]%	76.7 [58.8; 88.2]%	83.6 [71.2; 92.2]%	80 [63.1; 91.6]%
High inspiratory effort				
ΔCVP > 15 cmH_2_O	100 [66.1; 100]%	92.2 [83.8; 97.1]%	68.4 [43.3; 87.4]%	100 [92.5; 100]%

Data are expressed as the estimate of the diagnostic parameter [95% confidence interval]; CVP: central venous pressure; CI: confidence interval; PPV: positive predictive value; NPV: negative predictive value.

## Data Availability

The data presented in this study are available on request from the corresponding author. The data are not publicly available due to privacy and ethical restrictions.

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
