# Peer review of "Assessment of Inspiratory Effort in Spontaneously Breathing COVID-19 ARDS Patients Undergoing Helmet CPAP: A Comparison between Esophageal, Transdiaphragmatic and Central Venous Pressure Swing"

_diagnostics, 2023, doi:10.3390/diagnostics13111965_

Round 1

Reviewer 1 Report

I congratulate you for the idea you came up with in the assessment of respiratory effort, with obvious practical consequences. I liked the clear identification of the limits of the study, and I sincerely hope that you will continue the study, in order to consolidate the results obtained or to identify new indicators.

Author Response

We greatly appreciated the comment of the reviewer. A new study is planned to consolidate the results obtained.  

Reviewer 2 Report

The manuscript “Assessment of inspiratory effort in spontaneously breathing COVID-19 ARDS patients undergoing helmet CPAP: a comparison between esophageal, trans-diaphragmatic and central venous pressure swing. by Sergio Lassola et al. studies COVID-19 may be triggered by excessive inspiratory drive activation. However, there are several concerns with this manuscript:

Major:

1. The novelty is low, even though the authors repeated it with additional COVID-19 patients. Most of the main results of this manuscript have been published by authors ( PMID: 33635495).

Minor:

1. Some abbreviations should be given a full name when they first show up, eg. CPAP.

Reviewer 3 Report

I consider that although it is a small study due to the number of participants, it has an adequate statistical analysis and the minimum number to be considered a prudent analysis of the information, and reveals the usefulness of PVC compared to other variables, so provides relevant information that could later be used for a study with a larger number of participants

Despite what has been said, I consider it a relevant exploratory study worthy of publication.

Author Response

We thank the reviewer for this comment.

Our study has some limitations. In particular we studied a relatively small population, which is, however, comparable with that of other physiological studies. As the reviewer said, this is an exploratory study.

Round 2

Reviewer 2 Report

The novelty is low.